# Validation of Breast Cancer Margins by Tissue Spray Mass Spectrometry

**DOI:** 10.3390/ijms21124568

**Published:** 2020-06-26

**Authors:** Vitaliy V. Chagovets, Natalia L. Starodubtseva, Alisa O. Tokareva, Vladimir E. Frankevich, Valerii V. Rodionov, Vlada V. Kometova, Konstantin Chingin, Eugene N. Kukaev, Huanwen Chen, Gennady T. Sukhikh

**Affiliations:** 1National Medical Research Center for Obstetrics, Gynecology and Perinatology named after Academician V.I. Kulakov of the Ministry of Healthcare of Russian Federation, 117997 Moscow, Russia; n_starodubtseva@oparina4.ru (N.L.S.); alisa.tokareva@phystech.edu (A.O.T.); vfrankevich@gmail.com (V.E.F.); V_rodionov@oparina4.ru (V.V.R.); v_kometova@oparina4.ru (V.V.K.); bainat26@yandex.ru (G.T.S.); 2Moscow Institute of Physics and Technology, 141701 Moscow, Russia; kukaev@phystech.edu; 3V.L. Talrose Institute for Energy Problems of Chemical Physics, Russia Academy of Sciences, 119991 Moscow, Russia; 4Jiangxi Key Laboratory for Mass Spectrometry and Instrumentation, East China University of Technology, Nanchang 330013, China; chingin.k@hotmail.com (K.C.); chw8868@gmail.com (H.C.); 5Department of Obstetrics, Gynecology, Perinatology and Reproductology, First Moscow State Medical University named after I.M. Sechenov, 119991 Moscow, Russia

**Keywords:** direct mass spectrometry, lipidomics, breast cancer, tissue spray, molecular profiling, discriminant model

## Abstract

Current methods for the intraoperative determination of breast cancer margins commonly suffer from the insufficient accuracy, specificity and/or low speed of analysis, increasing the time and cost of operation as well the risk of cancer recurrence. The purpose of this study is to develop a method for the rapid and accurate determination of breast cancer margins using direct molecular profiling by mass spectrometry (MS). Direct molecular fingerprinting of tiny pieces of breast tissue (approximately 1 × 1 × 1 mm) is performed using a home-built tissue spray ionization source installed on a Maxis Impact quadrupole time-of-flight mass spectrometer (qTOF MS) (Bruker Daltonics, Hamburg, Germany). Statistical analysis of MS data from 50 samples of both normal and cancer tissue (from 25 patients) was performed using orthogonal projections onto latent structures discriminant analysis (OPLS-DA). Additionally, the results of OPLS classification of new 19 pieces of two tissue samples were compared with the results of histological analysis performed on the same tissues samples. The average time of analysis for one sample was about 5 min. Positive and negative ionization modes are used to provide complementary information and to find out the most informative method for a breast tissue classification. The analysis provides information on 11 lipid classes. OPLS-DA models are created for the classification of normal and cancer tissue based on the various datasets: All mass spectrometric peaks over 300 counts; peaks with a statistically significant difference of intensity determined by the Mann–Whitney U-test (*p* < 0.05); peaks identified as lipids; both identified and significantly different peaks. The highest values of Q2 have models built on all MS peaks and on significantly different peaks. While such models are useful for classification itself, they are of less value for building explanatory mechanisms of pathophysiology and providing a pathway analysis. Models based on identified peaks are preferable from this point of view. Results obtained by OPLS-DA classification of the tissue spray MS data of a new sample set (*n* = 19) revealed 100% sensitivity and specificity when compared to histological analysis, the “gold” standard for tissue classification. “All peaks” and “significantly different peaks” datasets in the positive ion mode were ideal for breast cancer tissue classification. Our results indicate the potential of tissue spray mass spectrometry for rapid, accurate and intraoperative diagnostics of breast cancer tissue as a means to reduce surgical intervention.

## 1. Introduction

Breast cancer ranks first in the incidence of malignant neoplasms among the female population. Currently, 60–80% of newly detected cases of breast cancer are treated with organ-preserving surgery [1,2]. Several large randomized trials clearly showed that there were no statistically significant differences in the rates of disease-free and overall survival among patients who underwent either mastectomy or organ-preserving operation [3,4,5]. At the same time, the risk of local recurrence after organ-preserving surgeries remains higher than after mastectomy, and is an average of 0.5% per year. One of the most significant factors of local recurrence is the status of surgical margins. Currently, the surgical margin is regarded as “positive” in the presence of a dye on an invasive tumor or ductal carcinoma in situ (DCIS). Preferably, the tumor is absent at more than 2 mm before the marginal edge of the resection. “Positive” margins of resection are the reason for performing repeated surgical interventions in 20–25% of breast cancer patients after performing organ-preserving surgeries [6,7]. Re-excisions are accompanied by poorer cosmetic results and dissatisfaction of patients, resulting in an increase in the cost of treatment. Therefore, it is extremely important to provide the surgeon with the most accurate information regarding the margins of resection during the operation and thereby reduce the risk of repeated surgical interventions.

The gold standard for evaluating the margins of resection is the morphological method of investigation. The sensitivity and specificity of histological and cytological methods are 73% and 98%, 89% and 92%, respectively [8,9]. The disadvantage of the morphological method is the need for enough time for its implementation. Urgent histological examination takes 20–40 min on average, and cytological—10 min. Conducting intraoperative ultrasound requires less time—3–6 min, but it is characterized by a lower sensitivity (75%) and specificity (81%) [10]. In addition, the accuracy of the method largely depends on the qualification of the specialist. Currently, digital radiography of remote samples is becoming increasingly popular. The sensitivity and specificity of this method are 83% and 95%, respectively [11]. The research takes only a few minutes. At the same time, it is necessary to state that not always the true dimensions of the tumor node coincide with the radiographic ones.

Dissatisfaction with both accuracy and time costs of existing methods of estimating the margins of resection forces the search for and development of new alternative approaches. One promising solution is the use of mass spectrometry (MS), which allows information on the molecular composition of samples and identifies tumor regions by the occurrence of specific proteins and metabolites. At the moment, several MS methods for the analysis of tissues have been developed, including mass spectrometry with rapid evaporation ionization mass spectrometry (REIMS) [12,13,14], desorption electrospray ionization (DESI) [15,16,17], matrix-activated laser desorption/ionization (MALDI) [18,19,20], secondary ion mass-spectrometric (SIMS) imaging [21,22,23], etc. Among these methods, REIMS has so far received the most recognition for the surgical MS analysis of tissues. On the basis of this method, an intraoperative method called the “intelligent knife” (iKnife) has been developed [12]. When using this method, tissue identification occurs during the operation in real-time by the characteristic profile of the mass spectrum of the tissue being cut [12]. However, the use of a “smart knife” is still not widespread, which is mainly due to its high cost and the need for a mass spectrometer in each operating room.

In our laboratory we are developing tissue spray mass spectrometry for the study of lipid markers of endometriosis tissues, lung cancer, and brain tumors [24,25,26,27,28,29,30,31]. The complete obviation of sample preparation and chromatographic separation stages allows rapid analysis without significant loss of chemical information [32].

In this study we tested the performance of the tissue spray mass spectrometry for the rapid and accurate differentiation between normal and tumor breast tissues.

## 2. Materials and Methods 

### 2.1. Tissue Handling

A study is performed on biopsy materials of breast cancer from patients treated at the National Medical Research Center for Obstetrics, Gynecology and Perinatology named after Academician V.I. Kulakov of the Ministry of Healthcare of Russian Federation (Moscow, Russia). All clinical investigations are conducted according to the principles expressed in the Declaration of Helsinki. All patients have read and signed informed consent approved by the Ethical Committee of the National Medical Research Center for Obstetrics, Gynecology and Perinatology named after Academician V.I. Kulakov. Biopsy samples of healthy tissues and malignant tumors separated by a histologist are taken from 25 patients for the development of a classification model. Samples of tissues with tumor–normal tissue borders analyzed by the histologist are taken for testing with the classification models. The samples are sliced for the histological study and the rest are frozen in liquid nitrogen and stored at −75 °C until the investigation. A small piece of a sample (approximately 1 × 1 × 1 mm) is cut, thawed and fixed on the needle in the ion source for tissue spray analysis.

### 2.2. Chemicals

Methanol and formic acid of HPLC grade were purchased from Sigma-Aldrich (St. Louis, MO, USA). Deionized water was purchased from Panreac (Barcelona, Spain).

### 2.3. Tissue Spray Mass Spectrometry of Breast Samples

MS analysis of tissue samples is performed on Maxis Impact qTOF (Bruker Daltonics, Bremen, Germany) with the in-lab designed ion source for tissue spray MS [28]. A mixture of H_2_O/methanol 1/9 with 0.1% formic acid is used for online tissue extraction and following spraying [30,31]. The solvent is supplied to the tissue with a flow rate of 1 mL/min by Dionex binary pump (Thermo Scientific, Germering, Germany). The potential of 3.8 kV is applied between a tissue and an inlet capillary in the positive ion mode and 3.1 kV in the negative ion mode. The distance between the sample and MS inlet is about 5–10 mm. Mass spectra are registered at a 2 Hz frequency resulting in 360 spectra for 3 min. The mass range is *m/z* 400–1000.

Tandem MS (MS/MS) is completed using data-dependent analysis with the following characteristics: the five most abundant peaks are chosen after full mass scan and are subjected to collision-induced dissociation, 35 eV collision energy, a 3 Da isolation window and 1 min of mass exclusion time.

### 2.4. Histological and Pathological Data

Microscopic samples are investigated by optical microscopy using an Olympus MX51 light microscope (Tokyo, Japan). The following characteristics of the biopsy are evaluated during histological investigation: the overall localization of the tumor and its localization by the quadrants of the breast, the borders of the tumor site, the maximum length and width of the node, the width of the node, histological type of cancer, the degree of malignancy, and the presence of metastatic lesions of regional lymph nodes. In addition to the standard histopathological study, all tumor samples are subjected to immunohistochemistry (IHC), which is used to determine the expression of estrogen (clone SP1, Ventana) and progesterone (clone 1E2, Venatana) receptors, proliferation index of Ki-67 (clone 30-9, Venatana) and her-2/neu protein expression (clone 4B5, Venatana). This hormone receptor status is graded using the Allred scoring and grading system. Separated cancer and healthy tissues identified by a histologist were used on the first stage of the investigation for statistical model training. During the second part of the study, tissues with normal regions, cancer regions and a margin in between were tested with the developed models. The results of the model classification were compared with the histological results.

### 2.5. Statistical Analysis of Mass Spectrometry Data and MS Peak Annotation

One hundred mass spectra were averaged over the stable total ion current (TIC) period and transformed into the abundance-m/z table for further analysis. Each peak abundance is normalized on TIC.

For multivariate analysis (MVA) data are normalized by Pareto scaling [33] prior to principal component analysis (PCA) and orthogonal projections onto latent structures discriminant analysis (OPLS-DA) [34], which are performed by in-house routines based on the *ropls* library [35]. The OPLS-DA is applied to the datasets in order to develop a sample classification model. The models are trained on MS data from 50 tissue samples. Three types of the MS data are used: all MS peaks with an intensity of over 300 counts; peaks with abundances which have statistically significant differences in abundances between normal and cancer tissue according to the Mann–Whitney test; and peaks identified as lipids. In addition, sets of peaks with variable influence on the projection of the model (VIP scores) higher than 1.0 are defined. In the case of identified features, they can be used as biomarkers and their biological meaning can be deduced.

The statistical significance of the ion’s abundance difference is determined by the Mann–Whitney test.

The lipids are annotated with in-lab-created R code (the RStudio version was 1.1.463 and the R language version was 3.5.2), which compares measured accurate *m/z* values with theoretical computer-generated values within 15 ppm. Sodium cation adducts and deprotonated molecules are considered in the positive ion and negative ion modes, respectively. More precise identification is completed based on the MS/MS data for the peak under consideration (Appendix A) if it underwent MS/MS analysis. Lipid nomenclature throughout the paper is in accordance with LIPID MAPS [36] terminology and shorthand notation summarized in [37].

### 2.6. Results and Discussion

#### 2.6.1. Tissue Spray MS Data

The data on a sample molecular composition are obtained by tissue spray mass spectrometry [24,25,26]. The positive and negative ionization modes are used to provide complementary information and to find out the most informative method for a breast tissue classification. The obtained mass spectra of the tissues contain 438 peaks over the threshold of 300 counts. The Mann–Whitney U-test reveals 152 peaks with statistically significant difference between normal and cancer samples. Of the 152 peaks, 64 are identified as lipids.

Characteristic mass spectra of tumors and surrounding tissues obtained in the positive and negative ion modes are shown in Figure 1. The most abundant peaks are observed in the m/z 600–900 mass range. These peaks correspond to different lipid species. Lipid peaks are typically dominated in tissue spray mass spectra [24,25,26], therefore, R-scripts generating theoretical lipid masses are developed to identify the obtained spectra. The possibility of ion formation by protons, sodium cation or potassium cation attachment in the positive ion mode is considered [29,30,31,32]. Deprotonation and chloride anion attachment are considered in the negative ion mode. Lipid identification is provided according to accurate mass within 15 ppm from the theoretical mass, and according to characteristic tandem mass spectra (Appendix A) [36,37]. The results of lipids identification are summarized in Appendix A for positive ions and in Appendix A for negative ions. Overall, 164 species are identified in the cancer and normal tissue extracts. These lipids belong to 10 subclasses, including phosphatidylcholines (PC), phosphatidylethanolamines (PE), sphingomyelins (SM), phosphatidic acids (PA), phosphatidylglycerols (PG), fatty acids (FA), phosphatidylinositols (PI), phosphatidylserines (PS), nonpolar glycerolipids (diacylglycerols (DG) and triacylglycerols (TG). The generated database contains masses of polar and nonpolar lipid classes with fatty acyls and alkyls varying from 10 to 26 carbon atoms and from 0 to 6 double bonds. Note that our results indicate that the use of both the positive and negative ion detection modes enhances the molecular coverage. Thus, the positive ion mode analysis provides information on phosphatidylcholines (PC), phosphatidylethanolamines (PE), sphingomyelins (SM), diacylglycerols (DG) and triacylglycerols (DG), whereas the negative ion mode analysis provides information on phosphatidic acids (PA), phosphatidylglycerols (PG), fatty acids (FA), phosphatidylinositols (PI), phosphatidylserines (PS) and phosphatidylethanolamines (PE).

#### 2.6.2. Classification Models Training

Semi-quantitative data on lipids are obtained using lipid peak abundance normalization to total ion current. The main differences (*p* < 0.05) in lipid level between normal and tumor tissues are found for three lipid classes: phosphatidylinositols (PI), phosphatidylcholines (PC) and sphingomyelins (SM) (Figure 2). Lipids within each class differ in attached fatty acyls, which are named in accordance with their total carbon number and double bond number (CN:DB).

PCA is performed to assess the experimental data quality. The PCA shows that the pooled quality control samples were clustered together (Appendix A), indicating that the MS analysis process met the required qualifications. Variances of 31% and 15% are described by the first and second principal components in the positive ion mode, respectively; variances of 27% and 17% are described by the first and second principal components in the negative ion mode, respectively. The PCA score plots demonstrate moderate clustering of data points corresponding to normal and tumor tissues without significant outliers.

Evaluation of the possibility of classification of tissues based on mass spectrometric data and identification of potential biomarkers of a breast tumor were carried out using the OPLS-DA multifactor analysis method. This method is a modification of the method of analyzing the main components, and its aim is to identify the differences between the studied groups using the set of samples with known diagnoses.

OPLS-DA models are created for the classification of normal and cancer tissue based on the following datasets: all mass spectrometric peaks over 300 counts; threshold peaks with a statistically significant difference of intensity determined by the Mann–Whitney U-test (*p* < 0.05); peaks identified as lipids; both identified and significantly different peaks. The parameters of the models under consideration are summarized in Table 1. R2 displays the proportion of data that the model describes using hidden variables. Q2 shows the expected accuracy of predicting new data.

The highest values of Q2 have models built on all MS peaks and on significantly different peaks. While such models are useful for classification itself, they are of less value for building explanatory mechanisms of pathophysiology and providing a pathway analysis. Models based on identified peaks are preferable from this point of view.

The same multivariate OPLS analysis of tissue spray MS data in the negative mode resulted in much worse classification models (Table 2). Even for the “all peaks” dataset, Q2 was only 0.643. We conclude that the analysis in the positive mode is better suited for rapid tissue classification.

#### 2.6.3. Classification Model Testing

Further validation of the OPLS-DA statistical model was completed by unsupervised classification of the tissue spray MS data on two histologically verified tissue samples containing both normal and cancer regions (Figure 3a,b) cut into 7 and 12 pieces, respectively. Several specific mass spectra from the sequential points are shown in Figure 1. Figure 3 presents a plot of tissue classification scores, calculated using previously built OPLS-DA models for the positive/negative ion modes and new MS data for particular pieces of tissue. Data points with a negative ordinate correspond to a normal tissue profile. Data points with a positive ordinate correspond to a cancer tissue MS profile. The bigger absolute values of the score are related to a more pronounced tissue type. The switch between negative and positive scores corresponds to a margin between normal and cancer regions.

Results obtained by OPLS-DA classification of the tissue spray MS data revealed 100% sensitivity and specificity (Table 3) when compared to histological analysis, the “gold” standard for tissue classification. “All peaks” and “significantly different peaks” datasets in the positive ion mode were ideal for breast cancer tissue classification. This result is in strict accordance with previous data obtained during the construction of OPLS-DA models. Moreover, the average analysis time for one tissue spot was about 5 min. This confirms the potential of using the tissue spray method for rapid and intraoperative diagnostics of breast cancer in order to reduce the surgical intervention.

## 3. Conclusions

Our results indicate that tissue spray mass spectrometry allows sufficient molecular specificity for the reliable differentiation between normal breast tissues and breast cancer tissues. Beneficially, the analysis of one tissue piece only requires 5 min and does not rely on the subjective assessment by a qualified clinician. Therefore, tissue spray mass spectrometry offers an alternative method for rapid and intraoperative determination of breast cancer margins.

## Figures and Tables

**Figure 1 ijms-21-04568-f001:**
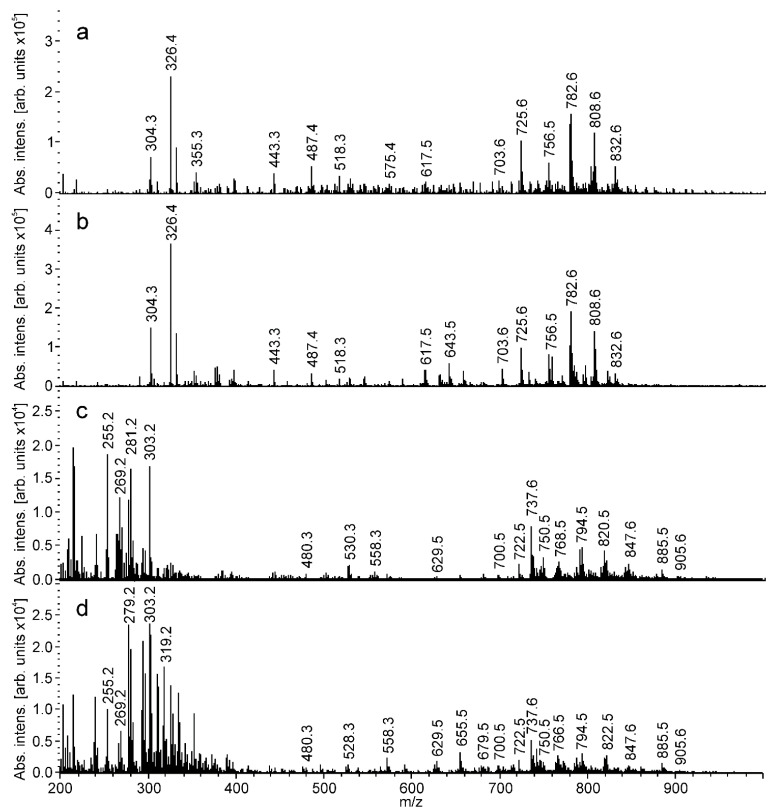
Characteristic tissue spray spectral profiles of breast samples recorded on Maxis Impact MS in *m/z* 200 to 1000. (**a**) Positive ion mode normal tissue; (**b**) positive ion mode tumor tissue; (**c**) negative ion mode normal tissue; (**d**) negative ion mode tumor tissue.

**Figure 2 ijms-21-04568-f002:**
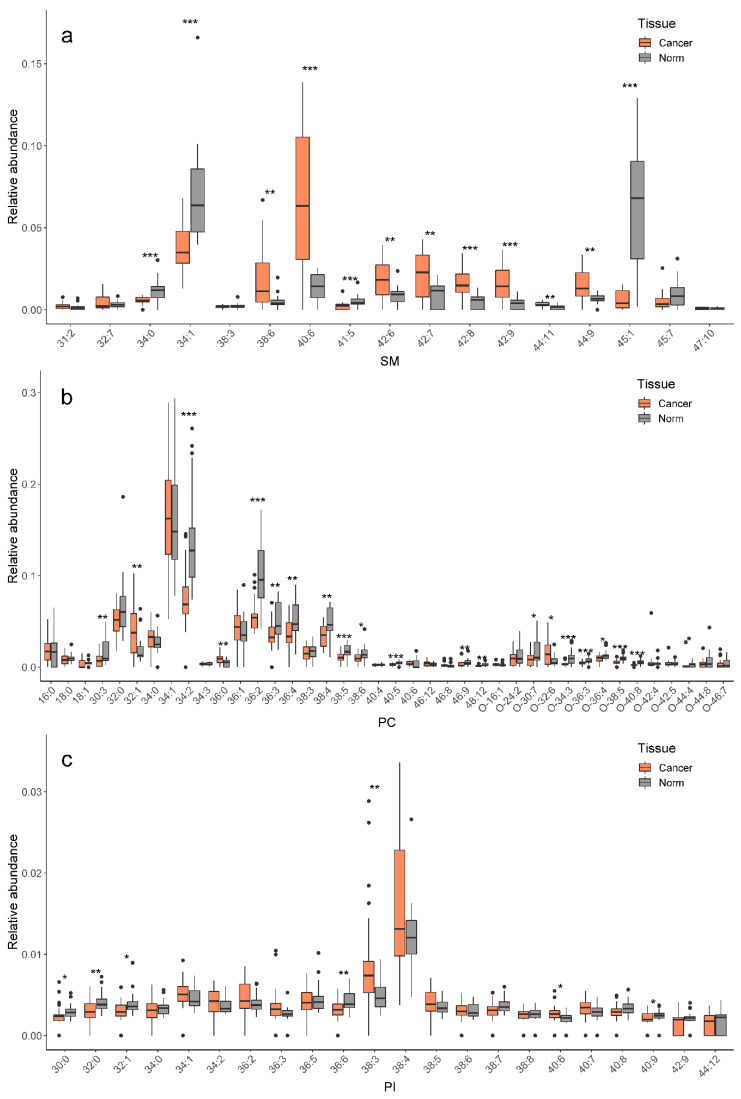
Relative abundances (%) of lipids in normal (gray) and tumor (orange) tissues: (**a**) SM, (**b**) PC, (**c**) PI. Lipid annotation: PI, phosphatidylinositols; PC, phosphatidylcholines; SM, sphingomyelins. SM and PC are detected in the positive ion mode, PI—in the negative mode. Statistically significant differences according to U-test are indicated by an asterisk: *—*p*-value < 0.05; **—*p*-value < 0.01; ***—*p*-value < 0.001. Black dots correspond to outliers.

**Figure 3 ijms-21-04568-f003:**
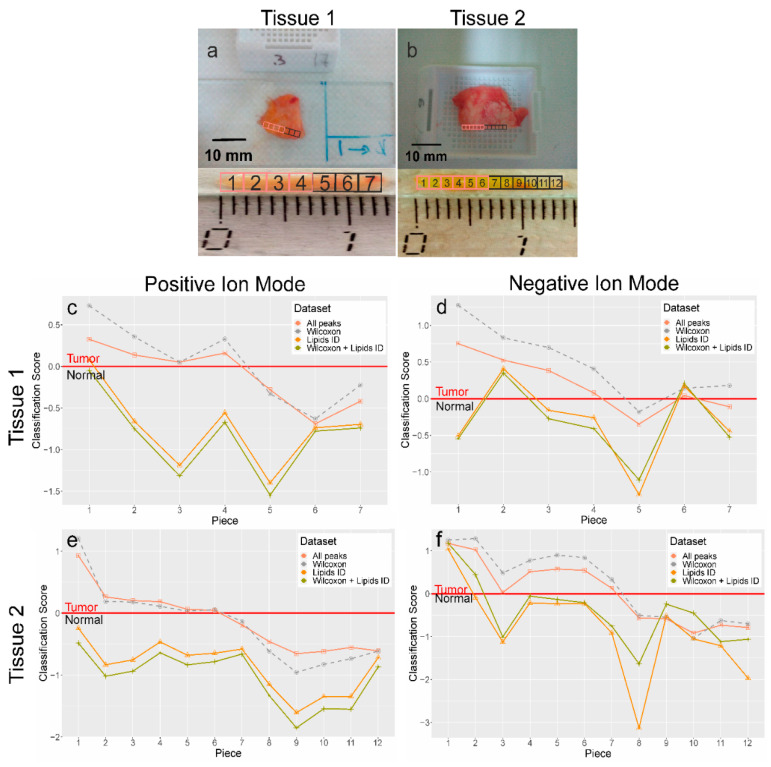
Validation of developed OPLS-DA models for tissue classification on a new set of samples (*n* = 19). (**a**,**b**)—The photo showing the pieces of two tissue samples that underwent both histological and tissue spray analysis. (**c**–**f**) The plot of tissue classification score vs. its spatial position for two samples for four types of datasets: all peaks, peaks with a statistically significant difference of intensity determined by the Mann–Whitney U-test, peaks identified as lipids, lipids with a statistically significant difference. Tissue spray MS is performed in the positive ion mode. The scores are obtained by unsupervised analysis of tissue spray mass spectra with the previously developed OPLS-DA models. The red line on the graph is determined by statistical model and separates the “normal region” from the “cancer region”.

**Table 1 ijms-21-04568-t001:** Summary of the data used for model development and parameters of the positive ion MS data of the OPLS-DA model. Statistically different peaks/lipids were considered at *p*-value < 0.05 according to the Mann–Whitney U-test with false discovery rate (FDR) correction.

Dataset		Model Parameters			
Name	Number of Variables	Number of Features with VIP > 1	R^2^X	R^2^Y	Q^2^
All peaks	541	102	0.438	0.868	0.829
Significantly different peaks	231	52	0.503	0.888	0.850
Identified lipids	106	22	0.512	0.845	0.784
Significantly different lipids	60	14	0.649	0.826	0.785

**Table 2 ijms-21-04568-t002:** Summary of the data used for model development and parameters of the negative ion MS data of the OPLS-DA model. Statistically different peaks/lipids were considered at *p*-value < 0.05 according to the Mann–Whitney U-test with FDR correction.

Dataset		Model Parameters			
Name	Number of Variables	Number of Features with VIP > 1	R^2^X	R^2^Y	Q^2^
All peaks	514	79	0.420	0.734	0.643
Significantly different peaks	190	36	0.490	0.753	0.579
Identified lipids	118	16	0.510	0.504	0.311
Significantly different lipids	40	7	0.706	0.479	0.381

**Table 3 ijms-21-04568-t003:** Summary on the data used for model development and parameters of the positive and negative ion MS data of the OPLS-DA model. Statistically different peaks/lipids were considered at *p*-value < 0.05 according to the Mann–Whitney U-test with FDR correction.

Tissue Sample	Dataset	Positive Polarity		Negative Polarity	
		Sensitivity	Specificity	Sensitivity	Specificity
1	All peaks	1.00	1.00	1.00	0.67
	Identified lipids	0.25	1.00	0.25	0.67
	Significantly different peaks	1.00	1.00	1.00	0.33
	Significantly different lipids	0.00	1.00	0.25	0.67
2	All peaks	1.00	1.00	1.00	0.83
	Identified lipid	0.00	1.00	0.17	1.00
	Significantly different peaks	1.00	1.00	1.00	0.83
	Significantly different lipids	0.00	1.00	0.33	1.00

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
