# Peer review of "Validation of Breast Cancer Margins by Tissue Spray Mass Spectrometry"

_ijms, 2020, doi:10.3390/ijms21124568_

Round 1

Reviewer 1 Report

Chagovets and coworkers reported the use of direct spray mass spectrometry to profile cancer and normal tissues in an attempt to differentiate the tumor tissues from the normal ones. Specifically, the authors used an in-house ion source for tissue spray to ionize compounds in the tissue by spraying a mixture of methanol, water and formic acid, and a Maxis Impact qTOF to record the mass spectrometric data. PCA and OPLSD were performed to analyze the data and generate statistical model for tissue classification. 100% sensitivity and specificity were achieved using data collected in positive ion mode, indicating the potential of MS in cancer tissue classification. Overall the manuscript is well written. But I have some comments and questions that need clarification.

Cancer and healthy tissues were used in this experiment. However, as the authors mentioned in the introduction, margin tissues are the major problem in cancer tissue removal as it is difficult to determine whether margins are health or “positive”. Have the authors analyzed any margin tissues and predict whether the samples are cancerous or normal and compared the results with the histological and cytological methods?

What the authors reported here (i.e., lipids as markers for cancer tissue classification) is not new. It has been reported previously by others. In addition to lipids, there might be some other compounds that can be used to differentiate cancer tissues from normal tissues. Did the authors identify any of them? For example, glycolysis is elevated in cancer cells. Did the authors observe any change/difference in the glycolysis pathway and/or other pathways particularly energy production pathway between cancer and normal cells?

The lipid identification appears to be based on a singular mass (Table S1) although MS/MS have been performed and the authors mentioned “More precise identification is done based on the MS/MS data for the peak under consideration, if it undergone MS/MS analysis”. Unlike FTMS or Orbitrap, qTOF data are not considered as accurate mass data. Identification based on a single m/z matching is not reliable in qTOF data and the authors need to provide MS/MS matching score and matched fragment ions.  

Author Response

Chagovets and coworkers reported the use of direct spray mass spectrometry to profile cancer and normal tissues in an attempt to differentiate the tumor tissues from the normal ones. Specifically, the authors used an in-house ion source for tissue spray to ionize compounds in the tissue by spraying a mixture of methanol, water and formic acid, and a Maxis Impact qTOF to record the mass spectrometric data. PCA and OPLSD were performed to analyze the data and generate statistical model for tissue classification. 100% sensitivity and specificity were achieved using data collected in positive ion mode, indicating the potential of MS in cancer tissue classification. Overall the manuscript is well written. But I have some comments and questions that need clarification.

 Cancer and healthy tissues were used in this experiment. However, as the authors mentioned in the introduction, margin tissues are the major problem in cancer tissue removal as it is difficult to determine whether margins are health or “positive”. Have the authors analyzed any margin tissues and predict whether the samples are cancerous or normal and compared the results with the histological and cytological methods?

  • The following description has been added to the manuscript to address the reviewer’s question: “Cancer and healthy tissues separated and identified by a histologist were used on the first stage of the investigation for statistical models training. During the second part of the study, tissues with normal, cancer regions and margin in between were tested with the developed models. The results of the model classification were compared with the histological results.” (Materials and Methods, Histological and pathological data, the bottom paragraph)  

What the authors reported here (i.e., lipids as markers for cancer tissue classification) is not new. It has been reported previously by others. In addition to lipids, there might be some other compounds that can be used to differentiate cancer tissues from normal tissues. Did the authors identify any of them? For example, glycolysis is elevated in cancer cells. Did the authors observe any change/difference in the glycolysis pathway and/or other pathways particularly energy production pathway between cancer and normal cells?

  • Indeed, the lipid biomarkers are not new in cancer tissue study, though tissue-spray for the breast cancer analysis was applied for the first time. Lipids as a data subset for a model development in the current study were selected because our previous investigation on comparison of tissue-spray and LC-MS demonstrated that the major signals in tissue-spray are those of lipids. An approach for lipids identification was developed during that study. As for other possible compounds, an additional investigation should be provided to compare conventional LC-MS and tissue-spray platforms to be sure exactly which compounds can be observed reliably. Other compounds are essential as the results of this investigation demonstrate good models with peaks different from lipids but they were not identified in the present work.

The lipid identification appears to be based on a singular mass (Table S1) although MS/MS have been performed and the authors mentioned “More precise identification is done based on the MS/MS data for the peak under consideration, if it undergone MS/MS analysis”. Unlike FTMS or Orbitrap, qTOF data are not considered as accurate mass data. Identification based on a single m/z matching is not reliable in qTOF data and the authors need to provide MS/MS matching score and matched fragment ions. 

  • The data on MS/MS identification were added to the manuscript. The data include the list of characteristic fragment ions for the classes under study (Table S1) and the column with identification rank in the Tables S2 and S3.

Reviewer 2 Report

The authors have done a pioneering job on the development of a potential novel clinical diagnostic approach of tissue spray mass spectrometry with 100% sensitivity, specificity and high efficiency (one tissue piece 5 minutes) . They have also demonstrated its application on differentiation of normal tissue and breast tumor tissues by OPLS-DA modeling. They had a rigorous design for experiment and data modeling. Here I would like to give some suggestions for future.

The systematic comparison for OPLS-DA modeling by using four different datasets was a great job to be appreciated and it provided a good reference for researchers using MVA mode.  Although lipid identification is not a major part for this paper, the identified lipids and significantly different lipids datasets have been used for OPLS-DA model development. As the authors presented in Result 3.1, only protonation, sodium cation and potassium cation attachment were considered for positive ion mode and deprotonation and chloride anion attachment were considered for negative ion mode,  this will probably affect the number of identified lipid as lipid molecule can be detected as other type of ion form by MS. Based on our lipidomics study for tumor sample cohorts, there are other major types of lipids having been identified from either positive or negative mode which were missing in the paper. e.g. in negative mode, such lipid subclasses as Cer, Hex1Cer, LPC  and phSM are always detected as [M+HCOO]-; in positive mode, such lipid subclass as TG is always detected as [M+NH4]+.  This may affect the OPLS-DA model development especially for the datasets of identified lipids and significantly-different lipids. It is also a key factor if you would go deep for diagnostic biomarker discovery as a lipid.  I suggest the authors to improve their in-lab lipid database capacity aiming for higher coverage of lipidome of tumor in future study.

Author Response

The authors have done a pioneering job on the development of a potential novel clinical diagnostic approach of tissue spray mass spectrometry with 100% sensitivity, specificity and high efficiency (one tissue piece 5 minutes) . They have also demonstrated its application on differentiation of normal tissue and breast tumor tissues by OPLS-DA modeling. They had a rigorous design for experiment and data modeling. Here I would like to give some suggestions for future.

The systematic comparison for OPLS-DA modeling by using four different datasets was a great job to be appreciated and it provided a good reference for researchers using MVA mode.  Although lipid identification is not a major part for this paper, the identified lipids and significantly different lipids datasets have been used for OPLS-DA model development. As the authors presented in Result 3.1, only protonation, sodium cation and potassium cation attachment were considered for positive ion mode and deprotonation and chloride anion attachment were considered for negative ion mode,  this will probably affect the number of identified lipid as lipid molecule can be detected as other type of ion form by MS. Based on our lipidomics study for tumor sample cohorts, there are other major types of lipids having been identified from either positive or negative mode which were missing in the paper. e.g. in negative mode, such lipid subclasses as Cer, Hex1Cer, LPC  and phSM are always detected as [M+HCOO]-; in positive mode, such lipid subclass as TG is always detected as [M+NH4]+.  This may affect the OPLS-DA model development especially for the datasets of identified lipids and significantly-different lipids. It is also a key factor if you would go deep for diagnostic biomarker discovery as a lipid.  I suggest the authors to improve their in-lab lipid database capacity aiming for higher coverage of lipidome of tumor in future study.

  • As we observed in our previous investigation on comparison of tissue-spray and LC-MS, the most abundant adducts in tissue-spray are protonated and sodiated molecules in positive ion mode and our unpublished data demonstrate that deprotonation and chloride anion attachment are the most abundant in negative ion mode, therefore this list of ions was chosen for the present study. The main issue preventing higher lipidome coverage is the same as its advantage - the absence of a separation stage. We will work to improve lipids coverage and we are grateful to the reviewer for the suggestions for future.